# The chiropractors' dilemma in caring for older patients with musculoskeletal complaints: Collaborate, integrate, coexist, or separate?

**Cecilia Bergström**[1]\*, **Iben Axén**[2,3], **Jonathan Field**[4], **Jan Hartvigsen**[5,6], **Monique van der Marck**[7], **Dave Newell**[8], **Sidney Rubinstein**[9], **Annemarie de Zoete**[9], **Margareta Persson**[10]

**1** Department of Clinical Sciences, Obstetrics and Gynecology, Umeå University, Umeå, Sweden, **2** Karolinska Institutet, Institute of Environmental Medicine, Stockholm, Sweden, **3** The Norwegian Chiropractors' Research Foundation "Et Liv i Bevegelse", Oslo, Norway, **4** Private Practice, Southsea, Hampshire, United Kingdom, **5** Center for Muscle and Joint Health, Department of Sports Science and Clinical Biomechanics, University of Southern Denmark, Odense, Denmark, **6** Chiropractic Knowledge Hub, Odense, Denmark, **7** Private Practice, Den Bosch, Netherlands, **8** AECC University College, Bournemouth, United Kingdom, **9** Department of Health Sciences, Faculty of Science and Amsterdam Movement Sciences Research Institute, Vrije Universiteit Amsterdam, Amsterdam, Netherlands, **10** Department of Nursing, Umeå University, Umeå, Sweden

\* cecilia.bergstrom@umu.se

**Data Availability Statement:** This research is based on qualitative data, making removing all identifying information difficult. Consequently, data

## Abstract

The world's elderly population is growing at a rapid pace. This has led to an increase in demand on the health and welfare systems due to age-related disorders, with musculoskeletal complaints driving the need for rehabilitation services. However, there are concerns about health services' ability to meet this demand. While chiropractic care is gaining recognition for its benefits in treating older adults with musculoskeletal disorders, there is limited scientific literature on chiropractors' role and experiences in this area. To bridge this gap, we interviewed 21 chiropractors in Great Britain, the Netherlands, Norway, and Sweden. Inductive qualitative content analysis was used to analyse the interviews, and despite differences in integration and regulation between the countries, several common facilitators and barriers in caring for and managing older patients with musculoskeletal complaints emerged. While participants expressed optimism about future collaborations with other healthcare professionals and the integration of chiropractic into national healthcare systems, they also highlighted significant concerns regarding the existing healthcare infrastructure. The participants also felt that chiropractors, with their non-surgical and holistic approach, were well-positioned to be the primary point of contact for older patients. However, there were some common barriers, such as the affordability of care, limited integration of chiropractic, and the need to prioritise musculoskeletal complaints within public healthcare. Our findings suggest that chiropractors experience their clinical competencies as an underutilised resource in the available healthcare systems and that they could contribute to and potentially reduce the escalating burden of musculoskeletal complaints and associated costs among older patients. Additionally, our findings highlight the desire among the participants to foster collaboration among healthcare professionals and integrate chiropractic into the national public

sharing beyond individual quotations in the manuscript is not feasible, as participants did not consent to publish their full transcripts. In addition, data sharing within this project is limited by Swedish confidentiality laws to guarantee the anonymity of the participating research subjects. Nevertheless, all pertinent data are comprehensively presented within the paper and its accompanying supplementary materials. A complete set of data used and analysed during the current study is available upon reasonable request from the Unit of Obstetrics and Gynecology, Department of Clinical Sciences, Umeå University (contact via Head of Unit Annika Idahl annika. idahl@umu.se), for researchers who meet the criteria for access to confidential data. However, inquiries for data access should first be sent to Cecilia Bergström (cecilia.bergstrom@umu.se).

**Funding:** This research was made possible by grants from the European Centre for Chiropractic Research Excellence (reference 42-2020-SE/CB) www.kiroviden.dk/eccre (CB) and Stiftelsen LKR Forskningsfond www.lkr.se/forskning/stiftelsen-lkr-forskningsfond/ (CB). The manuscript was prepared without involvement from the funders in study design, data collection and analysis, or decisions to publish. We acknowledge the support received from Umeå University (Umeå, Sweden).

**Competing interests:** The authors have declared that no competing interests exist.

**Abbreviations:** AHP, allied health professionals; BACE-C, BAck Complaints in Elders–Chiropractic; BACE-N, BAck Complaints in Elders–Chiropractic (Norway); BACE, BAck Complaints in Elders; GBD, Global Burden of Disease; GP, general practitioner; HRQoL, health-related quality of life; LBP, low back pain; MSK, musculoskeletal; QCA, qualitative content analysis; SMT, spinal manipulative therapy; WHO, World Health Organization.

healthcare system. Integrating chiropractors as allied health professionals was also perceived to improve coordinated, patient-centred healthcare for older adults.

## Background

The Global Burden of Disease (GBD) estimates that about 1.7 billion people globally are affected by musculoskeletal (MSK) conditions [1, 2], which are the primary reasons for disability and their related sequelae limiting people´s physical and mental health and affecting their workability and active social participation [3]. Today, older adults have better overall health, a better work environment and a higher education than previous generations [4]. Nevertheless, no reduction in the number of older individuals suffering from MSK complaints is found in the literature [1]. The prevalence of non-specific low back pain (LBP) appears to peak in the sixth decade of life [5], and the prevalence of back pain is predicted to rise sharply due to the ageing population worldwide [2, 5]. Additionally, these conditions demand rehabilitation and other healthcare services as they often co-exist with, or increase the risk of, other non-communicable diseases [6]. Yet older adults with MSK age-related disorders tend not to seek chiropractic care for their MSK complaints [7], and when they do, they receive less attention within the healthcare system and may be subjected to discrimination when seeking care compared to younger individuals with similar complaints [8].

Recent work has highlighted the need to understand the need to improve coordinated patient-centred healthcare for older adults, especially regarding MSK complaints [9]. Concerns about whether the current primary care model can meet the demands of all patients with MSK complaints and back pain exist [10]. The World Health Organization (WHO) has emphasised that integrating healthcare services will be crucial to maintaining functional ability among older adults to reduce the impact of the escalating demographic challenges [11]. Furthermore, the WHO recently issued updated recommendations for the non-surgical treatment and management of chronic primary LBP in adults within primary and community settings. The guideline has a special focus and recommendation for older adults and advocates for a holistic, person-centred, equitable, non-stigmatising, non-discriminatory, integrated and well-coordinated approach to patient care [12].

Chiropractors, who mainly see patients with MSK complaints, could act as primary contacts for older patients with LBP, among other professions, easing the pressure on the general practitioners (GPs) in the primary care setting [10]. Despite practice guidelines and the growing body of research showing the positive effects of chiropractic management for older adults with MSK complaints [13–18], older adults use chiropractic services to a lesser extent than the general population [19]. Identified barriers to not seeking chiropractic care among older adults with MSK complaints are lack of awareness of chiropractic, financial resources, logistics, and collaborative care [7, 20]. Nevertheless, scientific literature provides limited insights into the experiences and perspectives of chiropractors regarding the care and management of older adults with MSK complaints, which calls for further exploration.

### Aim

We aimed to explore chiropractors' experiences regarding facilitators and barriers in caring for and managing older patients (age 55+) with MSK complaints in primary care settings in four European countries.

## Material and methods

This study is part of the international prospective, multicentre cohort study of adults age 55 + with LBP in chiropractic care, the BAck Complaints in Elders–Chiropractic (BACE-C) study [21], which itself is modelled on the large international BACE study in primary care [22]. This BACE-C add-on is a two-phase project with qualitative and quantitative components undertaken in four European countries participating in the BACE-C (Great Britain, the Netherlands, and Sweden) or BACE-N (Norway) study. The countries were selected based on their involvement in the BACE-C and BACE-N (Norway) study. In this paper, we report on the qualitative components of this project.

### Design

We performed an inductive qualitative interview study with a purposeful sample of chiropractors in four European countries. The selected method allows for capturing the diverse and sometimes unexpected dimensions of different perspectives and is suitable for systematically exploring manifest and latent differences and similarities in data [23, 24].

### Setting

In Europe, chiropractic is a growing healthcare profession, serving as a primary contact for patients with MSK complaints [25]. Most chiropractors hold a 4- or 5-year master's degree and work in private practice with limited integration into the national healthcare systems [26]. However, such integration is heterogeneous across the countries included in the study, with only some chiropractors working as allied health professionals (AHP) [27]. In Great Britain, Norway, and Sweden (three of the study sites for our study), chiropractors are licensed healthcare professionals by each country's regulating body, while in the Netherlands (the fourth study site), chiropractic is considered 'alternative care'. Nevertheless, most chiropractors in the four included countries work in private practice with limited integration into their national public healthcare system. In Norway, the majority of chiropractors receive reimbursement for healthcare services from the National Insurance Scheme. In Sweden, certain county councils offer partial reimbursement, and others have subcontractor agreements with chiropractors. However, in Great Britain, chiropractic services are not widely accessible through the National Health Service (NHS). In the Netherlands, chiropractic care may be reimbursed through private healthcare insurance. Consequently, while public and insurance reimbursement schemes do exist, they are not widely available (with the exception of Norway), leaving the majority of patients in these countries to cover their expenses out of pocket.

### Recruitment and participants

We aimed to recruit a purposeful sample of twenty chiropractors, five from each of the four countries. These individuals were chosen based on their active involvement in either the BACE-C or BACE-N studies or known to provide care and, on a weekly basis, manage older patients (age 55+) with MSK complaints. Furthermore, all selected chiropractors were members of national chiropractic professional associations in Great Britain, the Netherlands, Norway, and Sweden. In total, we extended invitations for in-depth individual interviews to twenty-four chiropractors via email. However, four declined participation due to various reasons (Fig 1). In addition, the pilot interview (Swedish participant who fulfilled inclusion criteria and provided a signed participant's consent) was included in the analysis since no changes were made to the interview guide after the pilot interview. Consequently, the sample totalled 21 participants.

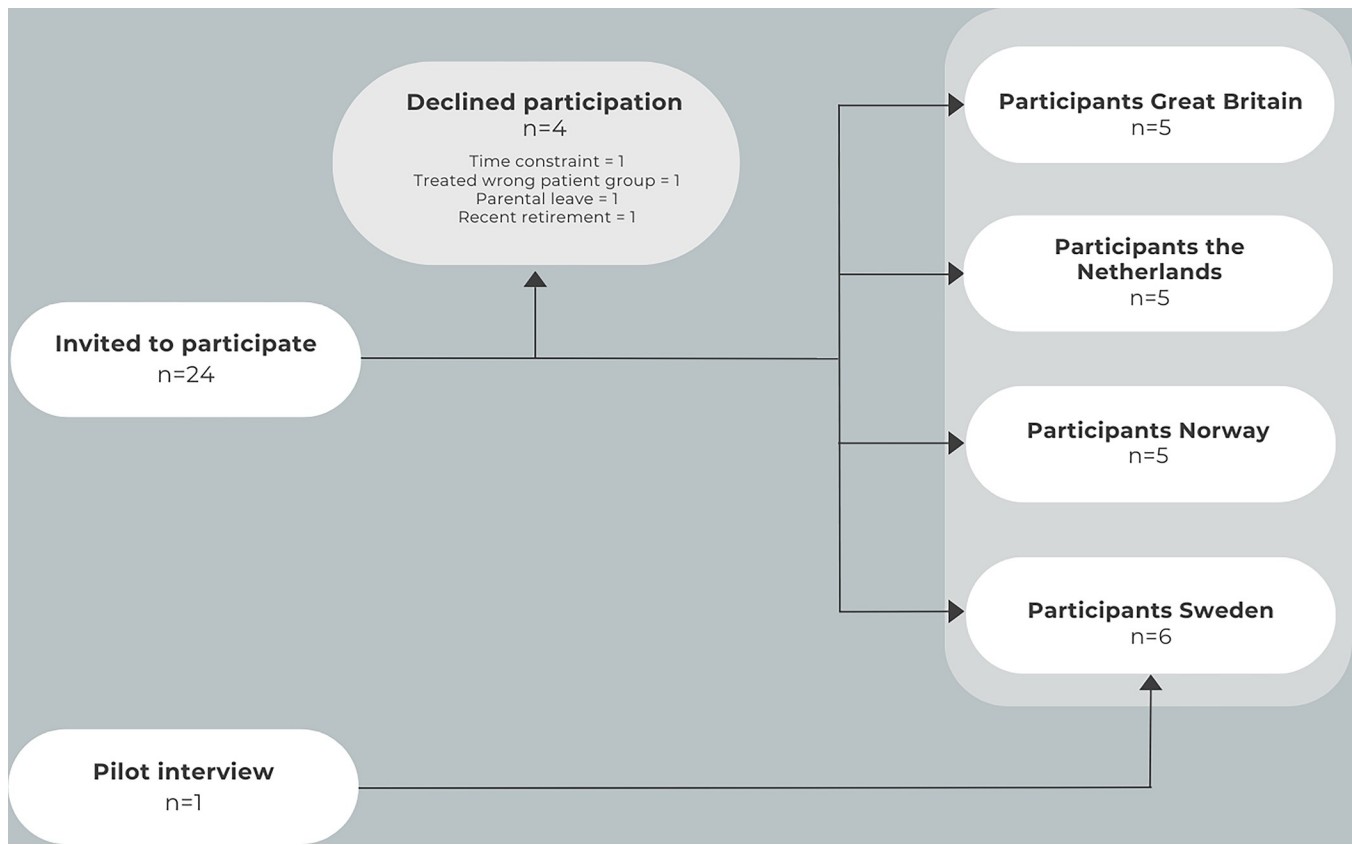

**Fig 1. Recruitment chart of the study group.**

## Ethics approval and consent to participate

Ethical approval was not deemed necessary by the Swedish Ethical Review Authority (reference 2021–04732). The project was approved in Great Britain via the AECC University College Research School of Chiropractic Research Panel (reference SOC-0422-014). No ethical approval was necessary in Norway or the Netherlands. All methods were performed in accordance with the relevant guidelines and regulations. Before the commencement of data collection, participants were provided with verbal and written information about the study. Both verbal and signed informed consent were obtained from all participants, including the pilot interview, before the interview started. Consent for publication was not required as the manuscript contains no personal information.

## Data collection

An interview guide was developed by the research team based on available literature and clinical experience. Before data collection started, a pilot interview was performed to evaluate the interview guide. During the pilot interview, close attention was paid to various factors, including the clarity and relevance of questions, the flow and structure of the interview guide, interview length and timing, rapport with the informant, quality of responses, and any challenges encountered. Feedback was collected from the informant after the interview, covering aspects such as the overall experience, question clarity, coverage of relevant topics, any confusion, redundancy, or irrelevant questions, comfort and interest levels, interview duration, and

recommendations for further improvement, resulting in no additional changes. As no changes were made to the interview guide, the pilot interview data was added to the data collection with the participant's signed consent. Data were collected by semi-structured interviews, also known as 'conversation with a purpose'; i.e., a reasonably informal or conversational nature that can be seen as a two-way dialogue [28]. Probing questions were posed throughout the interviews to enable participants to elaborate and develop their responses. A summary of the topics of the interview guide is presented in S1 Appendix. The mean length of the interviews was 63 minutes (SD 17), and the data resulted in an average of 135.6 coded segments per interview (SD 53.3).

Both verbal consent and a signed informed consent form were obtained from all participants. All the interviews (with only the participant and interviewer present) were conducted via Zoom® in an environment chosen by the participant, enabling flexible sessions across time zones without travelling. The pilot interview took place on November 17, 2021, with subsequent data collection conducted by the first author (CB female) from December 13, 2021, to July 13, 2022. Field notes were made directly after the interviews. The interviews were recorded with the participant's permission and then transcribed verbatim. All transcriptions were checked for accuracy.

## Qualitative analysis

Given the wealth of information collected during the in-depth individual interviews, we opted to split the material into two parts. The first part, focusing on integration, collaboration among health professionals, and organisational challenges, is presented in this paper. We applied a qualitative approach using qualitative content analysis (QCA) [23, 24] to explore chiropractors' experiences regarding facilitators and barriers in the care and management of older patients with MSK complaints. The transcriptions were analysed stepwise using QCA, according to Graneheim and Lundman [23, 24]. The selected analysis method is suitable for systematically exploring manifest and latent differences and similarities in data [23, 24]. The procedure of coding and developing categories was supported using the MAXQDA® software program [29]. As a first step, the transcriptions were read repeatedly to identify content areas. The emerging content areas were then discussed between authors CB and MP. Second, sentences and phrases containing relevant information related to the aim of the study were identified and condensed (i.e., the text was shortened without losing its meaning). The condensed meaning units were then labelled with codes, which were discussed repeatedly between authors CB (chiropractor and PhD) and MP (midwife and PhD with extensive experience in qualitative research) to reach a consensus. After that, the codes were compared for similarities and differences, forming clusters with similar codes within each cluster and different in content between clusters. Then, these clusters of codes were grouped into categories, reflecting the critical content of the interviews, i.e., the manifest part of the analysis. These emerging findings were presented to all co-authors for further input and validation. Finally, themes emerged, illustrating the interviews' latent substance (the underlying meaning running through the data), i.e., representing the latent phase of the analysis. To ensure credibility and trustworthiness, all authors discussed and assessed the coding and emerging findings to reach a consensus. The various authors' professions (chiropractors, a midwife, and molecular biologists) and extensive clinical experiences (i.e., authors' pre-understanding) further strengthened the confirmability of the analysis.

## Results

### Participants

The participants in this study typically handle a self-estimated patient load ranging from 20% to 80% of older patients with musculoskeletal complaints per week. The majority of

**Table 1. Demographic overview of the study sample.**

| Variable | All participants |
|---|---|
|  | *N = 21* |
|  | *n (%)* |
| **Gender** |  |
| Male | 9 (42.9) |
| Female | 12 (57.1) |
| **Age** |  |
| Mean; SD[a] | 45.3; 11.5 |
| Min—Max | 29–61 |
| **Country** |  |
| Great Britain | 5 (23.8) |
| Netherlands | 5 (23.8) |
| Norway | 5 (23.8) |
| Sweden | 6 (28.6) |
| **Time in days between ICF sent and received** |  |
| Mean; SD[a] | 14.3; 20.8 |
| Min—Max | 0–73 |
| **University** |  |
| Ango-European Chiropractic College | 13 (61.9) |
| Northwestern College of Chiropractic | 1 (4.8) |
| Palmer College of Chiropractic | 2 (9.5) |
| University of Southern Denmark | 2 (9.5) |
| University of South Wales | 2 (9.5) |
| University of Surrey | 1 (4.8) |
| **Years in chiropractic practice** |  |
| Mean; SD[a] | 20.1; 11.8 |
| Min—Max | 3–38 |

[a] Standard deviation

participants were female (n = 12, 57%), and most (n = 13, 62%) had completed their education at the Anglo-European College of Chiropractic (based in Great Britain), with a mean experience of 20.1 (SD 11.8) years in chiropractic practice. None of the participants had higher education beyond their chiropractic degree. All participating chiropractors worked in a private setting; however, two Swedish chiropractors worked in clinics with subcontractor agreements with the public healthcare sector, while all participating Norwegian chiropractors are partially reimbursed for healthcare services provided by the National Insurance Scheme. Some practitioners had prior subcontractor agreements with the public healthcare sector. A demographic overview of the study sample is presented in Table 1.

The overarching theme emerging, "Collaborating, integrating, co-existing or being separated when treating older patients with musculoskeletal complaints", serves to illustrate chiropractors' experiences of providing care and management to this group of older patients. Most participants in this study envisioned seamless collaborations among healthcare professionals or the integration of chiropractors into the nationally funded healthcare systems. However, there was a contrasting perception that chiropractors also could either co-exist alongside these healthcare systems or even be disregarded by them entirely. In all four countries, shared hopes and concerns emerged regarding the role of chiropractors in addressing the needs of a sizable

**Table 2. Illustrative overview of the findings.**

| Overarching theme | Sub-themes | Categories |
|---|---|---|
| Collaborating, integrating, co-existing or being separated when treating older patients with musculoskeletal complaints | Having a place and a role to fill within the healthcare system | Providing timely and effective treatment for older patients |
| | | Filling a massive gap in primary healthcare |
| | Facing the challenges of diverging patient groups | Focusing on preventive treatments |
| | | Providing care to patients with high resources |
| | | Struggling to aid patients with low resources |
| | Valuing the pros and cons in the different healthcare organisations | Balancing the strengths and shortcomings when working in the public sector |
| | | Facing opportunities and obstacles working in the private sector |

patient group of older adults who often face challenges in accessing timely care and management within the existing healthcare framework.

Several common barriers and shortcomings are experienced, irrespective of whether working in the public or private healthcare sector or a specific country. For example, the participants expressed a lack of subsidised care for patients with low financial resources, insufficient funding, and staff shortages to meet the care needs of the ever-growing population of older adults with MSK complaints. In addition, lack of acceptance, collaboration and insufficient communication between different healthcare professionals and care settings was viewed as a potential patient safety issue. Facilitating factors included a perceived high competence in assessing and managing older patients with MSK complaints, which was mostly underused in public healthcare. On the one hand, some participants expressed appreciation for the freedom of working in private care. On the other hand, other participants preferred teamwork and collaborating with other professionals in the public sector. The shared opinion was that chiropractors could contribute significantly to society and medical services through improved care and management, decreasing expenditure, and the ever-increasing pressure on the public healthcare system if participation as AHPs were introduced more widely.

As follows, the sub-themes are presented with a short introduction summary followed by a presentation of the content of each category, illustrated by quotes from the participants. Table 2 gives an illustrative overview of the findings.

**1. Having a place and role to fill within the healthcare system.** Many participants believed that MSK complaints in older patients received too little attention and that chiropractors could safely and effectively assess and manage older patients with MSK complaints. Chiropractic care and management were thought to have a positive socioeconomic impact on individual and societal levels. This sub-theme consists of two categories presented below: "Providing timely and effective treatment for older patients" and "Filling a substantive gap in primary healthcare".

*1.1 Providing timely and effective treatment for older patients.* Participants viewed chiropractors as highly competent and well-positioned to help older patients with MSK complaints. Furthermore, comments suggested a view that chiropractors' competence and management strategies could offer older patients better and more effective help for their MSK complaints than other healthcare professionals.

> Chiropractors have a place in musculoskeletal healthcare as specialists. We have more knowledge than the average GP about these problems and are better suited to send a patient in the right direction and look at what kind of care this patient needs. (Interview 10)

Chiropractors working in private care could generally schedule appointments for older patients for their MSK complaints shortly after the initial contact. The general view was that the referral rate of older patients with MSK complaints to chiropractors would increase if chiropractors were better integrated into the public healthcare system. However, participants expressed a belief that healthcare professionals in the public healthcare system were reluctant to refer or recommend older patients with MSK complaints to someone outside the public healthcare system unless there was a personal contact established, leaving the patients waiting for treatment or limited to other forms of treatment.

> We developed relationships with some of the consultants, and they will, from time to time, refer. (Interview 19)

*1.2 Filling a substantive gap in primary healthcare.* Chiropractors described that they were in a unique position to help older patients maintain their health and positively influence their health-related quality of life (HRQoL) by assisting them to regain or better manage what had been lost due to pain and disability. With a holistic patient care approach to patient care, including but not limited to manual therapy, rehabilitation exercises, and psychosocial support, the experience among participants was that chiropractic could offer much more than just pain relief to older patients, i.e., enable the older patient to maintain mobility, thus reducing inactivity and potentially deterioration. In addition, many participants acknowledged a knowledge gap in the assessment of MSK complaints that needed to be addressed in the current public healthcare model, which was predominantly occupied by GPs and physiotherapists.

> Chiropractors have a unique position to help people maintain their health and a good quality of life and regain it if they lost it. It doesn't just come down to manipulation. It comes down to many techniques, good communication, and showing them how to be active [. . .]. We're in a unique position to help the ageing population. (Interview 21)

In various ways, all participants expressed that MSK complaints needed more attention in the public healthcare arena to benefit older patients. However, there was an understanding that other more severe health conditions, i.e., cardiovascular disease and diabetes, as well as mental health issues, were rightfully prioritised, thus taking up a large proportion of time and resources. Nevertheless, the view was that if patients experienced less pain, disability, and better physical function, it would positively affect autonomy, HRQoL, and other more severe health conditions, including mental health. Further, they expressed that little was done or was available for older patients with MSK complaints in the public sector and that there was not enough funding for appropriate rehabilitation equipment regarding the rehabilitation of MSK conditions. In short, participants agreed that more must be done to enable older adults to maintain function and autonomy long after retirement.

> For the sake of the patients, we need to raise awareness of the musculoskeletal field within healthcare services. The [patient] group with heart and lung issues and diabetes are large and heavy groups taking up most of the healthcare resources. If they [the patients] experienced better function, it would have a positive effect on other problems [cardiovascular diseases and diabetes] as well, including mental health issues. (Interview 6)

Concerns were raised about whether there would be enough chiropractors to fill the identified gap regarding the care and management of these patients within public primary healthcare in addition to meeting the demand from the growing population of older adults with MSK

complaints. Some participants also mentioned that it might be challenging to convince chiropractic clinicians to leave their private practice and work within the current public primary healthcare model.

> There have been attempts to accept patients from the National Health Service, and our clinic tried that out a few years ago. There was not enough space in the diary to accommodate the patients [. . .], and we were very busy. That became a stress over time; we couldn't meet the demand. (Interview 21)

Chiropractic care, offering a holistic care approach, including various treatment and management strategies tailored to patient preferences and needs, was seen as an underutilised resource in the care and management of older patients with MSK complaints. The engagement of chiropractors as primary contact for older patients was believed to reduce the waitlist and GPs' workload, thus freeing up time for the GPs to attend to other, more severe conditions.

> We are very qualified to deal with the MSK aspects because many conditions we've been seeing going through GPs are discogenic or mechanical low back pain, hip impingements, and tension-type headaches. All things that we can manage are taking the workload off the GPs and getting the patients back to work very quickly. (Interview 18)

**2. Facing the challenges of diverging patient groups.** Participants believed preventive care could increase patients' HRQoL, decrease sick leave/disability pension, improve function and autonomy, and reduce lifestyle-related diseases, i.e., obesity, diabetes, and cardiovascular diseases in this patient group. Identified challenges were access to affordable care independent of personal financial resources and the area lived in. Still, challenges to providing needed care were identified and presented in three categories: "Focusing on preventive treatment", "Providing care to patients with high resources", and "Struggling to aid patients with low resources".

*2.1 Focusing on preventive treatments.* Participants believed that older patients may have an increased risk of being on sick leave due to MSK complaints, decreasing the chances of returning to work and increasing the risk of persistence of symptoms. Preventive or early treatment of MSK complaints was seen as critical to increase patients' HRQoL and individual autonomy and reduce lifestyle-related diseases, sick leave, disability pension and thus societal costs.

> We tend not to think about the long-term consequences, even though someone at age 75 might live to 100. That's still 25 years, so long-term thinking disappears a little. (Interview 18)

Public expenditures on preventive care needed to be increased; however, preventive care was not seen as a prioritised area in the current public primary healthcare model. Participants expressed that innovative prevention strategies and public health campaigns targeting older adults focusing on lifestyle choices and physical activity were necessary to reduce lifestyle-related disorders and lower healthcare costs.

> I think that healthcare could be a little bit more innovative [. . .] if the focus is on decreasing the cost of healthcare; there is a lot to be gained from working preventively. (Interview 10)

*2.2 Providing care to patients with high resources.* Most patients in the age 55+ group who were seeing a chiropractor for their MSK complaints were considered to have a higher

educational level, better personal finances and generally belong to the middle or upper socio-economic demographic section. Some patients were still working professionals, but most were retired, wanting to continue being physically and socially active. However, depending on the catchment area of clinical practice, some participants predominantly proved care for labourers, farmers, office workers, and patients from female-dominated occupations, i.e., care assistants, cleaners, and nurses. Patients in manual labour jobs often sought care to enable them to continue to work. However, resourceful patients were generally perceived as more likely to receive care and adhere to treatment plans.

> When you go up in age, it tends to be the ones who are better off because they can afford their care [. . .] they are aware that there are things that they want to keep doing and continue to lead an active lifestyle, and their pain inhibits them from doing so. (Interview 16)

*2.3 Struggling to aid older patients with low resources*. Participants perceived an apparent inequality in access to care based on the patient's financial ability to pay. Low educational level was also considered a barrier to chiropractic care and management. Many participants also stated that some older patients were often scheduled for longer appointments to accommodate their needs but were rarely charged accordingly. Despite being unable to afford care, some patients still came in for treatment, however more sparingly. Some participants mentioned that they adapted their prices for patients with poor financial resources to enable continued care.

> In the long term, if the patient just doesn't have the money, we adapted our prices. (Interview 18)

Subsidised care was seen as something positive, both from a patient and clinician's point of view. However, treatment free of charge could be abused by patients. Participants believed it might decrease patients' initiative and responsibility in the care and management of their MSK complaints, making it more challenging to use a more active treatment approach. Administration linked to reimbursement was also considered time-consuming and needed to be simplified.

> If they know it's going to be expensive and that it's going to take some time, they need to take the initiative. They know that they can't come in because it's cosy. It is a clear advantage for us that they must take the initiative. . . (Interview 5)

Patients who struggle financially were viewed as a more challenging patient group since they often seek help at a later stage, meaning they were generally worse off and often presented with persistent symptoms, multiple issues, poor general health, several lifestyle-related diseases, and being on sick leave, thus having a worse prognosis. Some patients had an immigration background [30], which added to the challenge of navigating the healthcare system and accessing necessary medical services. Therefore, there was an apparent need for subsidised chiropractic care for some patients with poor financial resources.

> The patient's financial situation is the barrier to what actually can be done to a high degree. (Interview 6)

**3. Valuing the pros and cons in the different healthcare organisations.** The sub-theme "Valuing the pros and cons in the different healthcare organisations" describes the

participants' perceived advantages and disadvantages of working in public or private health-care, irrespective of their workplace. This sub-theme contains the categories: "Balancing the strengths and shortcomings when working in the public sector", and "Facing the opportunities and obstacles working in the private sector".

*3.1 Balancing the strengths and shortcomings when working in the public sector.* Most responders desired to be part of a larger social context, i.e., public primary healthcare, for the patients' and personal benefits. Working within the public healthcare system provided some security regarding employment and benefits, including paid sick leave and vacation.

> I would like to work in public healthcare. There are many advantages to not just working as a self-employed person. Financially, including sickness benefits, the security that comes with that. (Interview 5)

However, this option was impossible for some participants as chiropractors within public primary healthcare did not exist in their country. At the same time, many participants expressed some ambiguity regarding working in public primary healthcare. The participants said that primary healthcare in the public sector was underfunded and overstretched, resulting in a hostile work environment. Further, the lack of personnel resulted in long waiting lists for older patients to be assessed for their MSK complaints, thus increasing the risk of more persistent problems. The growing number of older adults was seen as a major challenge for the healthcare system by most participants regardless of country of practice, and the lack of family physicians (resulting in patients always seeing a new doctor) for each patient made the public healthcare system seem vulnerable, compromising patient safety.

> I think the role of the family physician, the local community physician, has been lost, and I don't think GPs have the time or the resources these days to get to know their patients, understand the biopsychosocial issues behind them, and manage their complaint accordingly, taking everything into account. (Interview 17)

Participants also expressed concerns about resistance from other healthcare providers allowing chiropractors to fill the role of primary contact in public primary healthcare for this group of patients. Despite these concerns, participants still emphasised the potential benefits of integrating chiropractors into the public healthcare system to meet current and future demands from older patients with MSK complaints.

> It is difficult for a chiropractor to fit into the role at a primary care centre. The chiropractor is educated to lead and make decisions and is quick at that [. . .] it will be difficult to fill that void that you would want to fill because I don't think that anyone would let you [. . .] (Interview 3)

*3.2 Facing the opportunities and obstacles working in the private sector.* Working in the private sector was expressed as coming with a significant sense of freedom in choosing how to work within the national legislative expectations and regulator's code of conduct. A healthy work environment and the clinician's well-being were seen as essential to delivering good care. The atmosphere in a private setting was often perceived as more comforting and less stressful for patients than in public healthcare, which might positively influence recovery. Nevertheless, the perception was that better integration would benefit the chiropractic profession but not necessarily be better for the patient and the chiropractor.

[. . .] This is not like a hospital setting; it's a different atmosphere. I'm unsure if I want to take that atmosphere [in the clinic] away from the patient because it's relaxing in many ways. (Interview 2)

On the downside, becoming isolated as a professional in a private setting was mentioned. An isolated work situation was perceived as a potential risk of falling into the use of potentially questionable methods to attract patients or misusing reimbursement from third parties. Working with other healthcare professionals from different disciplines and exposure to other ideas and ways of thinking was considered beneficial for continued professional education, decreasing the risk of unethical approaches and improving patient safety and satisfaction.

If you're part of a GP practice, you can talk to the GP before you see a patient and ask about the medication and how it works. That way, you're discussing with someone else, and not just with the patient, someone who knows about it [the medical condition]. You are more engaged with this piece of knowledge. (Interview 14)

Access to critical health information of older patients was perceived as limited in private settings. Most participants experienced challenges regarding collaboration with other healthcare providers in the public healthcare system. Better and more accessible communication with other healthcare professionals was frequently mentioned to increase patient safety. Communication and collaboration with other healthcare professionals were often developed over time and were primarily based on personal contacts. Referral procedures were complicated, tedious, unnecessarily long and did not generate an income for the chiropractor in private practice. Yet, most participants viewed communication as an essential cornerstone of quality patient care and a way of improving the chiropractic profession's reputation among other healthcare professionals.

I don't like the isolationist clinician that I am. It's not good for bridge-building in the profession. There are quite a few barriers to it as well. Still, we need to write and communicate with the healthcare community, keeping them informed from an individual clinician level and then bridges will slowly build. We may get more collaboration between the healthcare professions at a higher interdisciplinary level. (Interview 17)

## Discussion

Despite variations in the integration and regulation of chiropractic between Great Britain, the Netherlands, Norway, and Sweden, our study revealed several similarities regarding the facilitators and barriers in caring for older patients with MSK complaints. These shared views and experiences underscore a clinical dilemma faced by the participants as they navigate their roles as part of the patient's healthcare team yet not officially integrated into the wider healthcare systems. Although most participants envisioned a better-integrated future, they also expressed concerns regarding the existing healthcare systems. In their opinion, these healthcare systems were perceived to suffer from challenges such as inadequate funding, staffing shortages, and deficiencies regarding interdisciplinary collaboration and communication. Our findings shed light on significant organisational problems that are perceived to exclude both professionals and patients from accessing nationally funded care, particularly affecting resource-challenged patients or those with limited financial resources. The recognition of these barriers underscores the importance of adaptability and strategic decision-making while simultaneously emphasising a need to address these systemic issues to ensure equitable access to care.

Chiropractic has a long history of a holistic (framed contemporality as biopsychosocial) approach to patient care [31, 32], an approach also appreciated by care recipients [7, 33]. Besides manual therapy, contemporary chiropractors often use several other modalities to care for and manage MSK complaints [20, 32], including providing health promotion advice [33], adapting the treatment to patient preferences and needs and taking into account the whole therapeutic encounter as a complex web of mental as well as physical processes that impact MSK pain [34], which aligns well with recommendations by the WHO [12]. However, there are several complicating factors in the care and management of older adults, such as co-morbidities and chronic underlying diseases, i.e., cardiovascular disease, diabetes, neurological disease, and osteoporosis [32]. Other life-long age-related conditions include, but are not limited to, hearing loss, declined vision, poor balance, cognitive impairment, mobility, and psychosocial issues [35], making the care and management more challenging not only for the individual clinician but also posing challenges for the health and social welfare systems at large [5, 8, 36, 37]. Previous research has shown that patients in general practice typically exhibit higher rates of comorbidities, experience longer episodes of LBP, have more sick leave, report poorer HRQoL, and display more negative psychosocial factors than patients who seek chiropractic care [38–40]. Furthermore, patients receiving chiropractic care tend to exhibit better physical and social functioning and are less likely to belong to lower socioeconomic categories [40]. Nevertheless, participants in our study believed that chiropractors were highly qualified and better suited than other healthcare professionals to assess, manage, and provide more than just pain relief to older patients with MSK complaints. This perception among participants may arise because chiropractors typically hold a 4- or 5-year master's degree, whereas physiotherapists typically complete a 3-year bachelor's degree. Furthermore, there is evidence indicating that GPs may need to improve their understanding and management of LBP to meet international evidence-based standards [40]. Furthermore, the participants' average years in practice surpass 20 years. It is plausible that with such substantial accumulated clinical experience, participants perceive themselves as more competent in managing complex cases within this area compared to other healthcare professionals, and such perceptions contribute to their confidence in addressing complex patient needs. However, these perceptions may very well be unrealistic, and further exploration is warranted to assess the extent to which this perceived competence aligns with objective measures of clinical proficiency and patient outcomes.

Patients with MSK conditions experience increased health expenditures related to the number of visits to healthcare professionals, medications, imaging, injections, and surgery [41–46]. Over time, most patients stop consulting their GP or physiotherapist despite persistent or recurrent symptoms and considerable pain medication consumption [47]. Dissatisfaction and perceived ineffectiveness with the care offered by GPs and physiotherapists appear to be the prime facilitators for most patients to seek chiropractic care for their MSK complaints [7, 33]. However, retired older adults suffering from multiple co-morbidities generally cannot afford private care, even with insurance co-pays [7, 20]. Nevertheless, the positive benefits of treatment may decrease the concern about cost by some [33]. Participants in our study perceived financially struggling patients as having poorer general health and prognosis and conveyed that many older patients lacked the financial resources to get preventive or early care within the private sector, leading to unequal access to needed care, as shown by other studies [7, 20, 33]. Furthermore, the cost was perceived to deter some older patients from adhering to treatment plans warranting subsidised chiropractic care. While participants in our study generally viewed subsidised care positively, the challenge of balancing affordability with patient motivation to engage in health management was evident. Personalised patient education materials, strong patient-provider communication and trust can enhance health literacy and empower patients to play a more proactive role in their care [48]. Additionally, community-based group

exercise initiatives and partnerships with local organisations offer promising avenues enabling older adults to engage in regular, suitable, and health-enhancing physical activity while fostering social interaction [49]. Although the integration of chiropractic into mainstream healthcare and associated subsidies remain patchy or absent, the chiropractic clinician must evaluate the most effective treatment approach while being sensitive to patient preferences and needs to reduce costs and improve patient outcomes and satisfaction.

Chiropractic care delivered in an environment imbued by patient-centeredness and an active approach to patient care and management has shown to be an effective alternative for managing musculoskeletal disorders [50, 51]. Studies from the US and UK have found that patients also have a positive attitude towards chiropractic as a primary intervention for LBP and consider chiropractic care an alternative to surgery and pain medication [20, 33]. Apart from timely and preventive care, flexibility in scheduling, accessibility, and adapting consultation lengths were seen by the participants in this study as essential facilitators to chiropractic care of older adults with MSK complaints. These findings are consistent with the results from a recent study from the Netherlands [7]. In line with our research, older patients who sought chiropractic or medical care for their MSK complaints in the US were found to rarely be co-managed or receive concurrent care [20]. Furthermore, most participants in our study found other healthcare professionals reluctant to refer or recommend older patients with MSK complaints to chiropractors. This aligns with prior research findings indicating that primary care physicians generally prefer patients to contact chiropractors independently rather than initiate formal referrals themselves [52]. The hesitancy among healthcare professionals to issue formal referrals may have implications for the efficiency, quality, and safety of patient care within the healthcare system, possibly leading to disruptions in the continuity of care. Because patients mostly pay for chiropractic care out of their own pockets, this may deter both healthcare professionals and patients from pursuing chiropractic treatment options, contributing to the observed reluctance in referrals.

A positive and healthy work environment and the clinicians' well-being were essential to the participants' day-to-day clinical practice, irrespective of the participant's workplace. A healthy psychosocial work environment is imperative in attracting and retaining healthcare professionals [53, 54] and is associated with good patient care and outcomes [55]. Studies among healthcare professionals found daily work-related stressors, i.e., ineffective work processes, excessive workload, low support, and decreased autonomy [56, 57], negatively associated with a person's well-being [56–58]. Despite several advantages associated with private practice and ambiguity towards working within public healthcare, the majority opinion was a desire for chiropractic to become more integrated into mainstream healthcare for the benefit of the older patient and the chiropractic profession.

Historically, communication and collaboration between chiropractors and other healthcare professionals have not been ideal [59], often overshadowed by scepticism, mistrust, or a belief that chiropractic care is ineffective for musculoskeletal complaints [60]. However, over the years, chiropractic has enjoyed greater acceptability [26], and even though interprofessional practice needs to be enhanced, intra-clinical therapeutic alliances have improved [61]. Good communication, collaboration, continuity of care and record sharing between chiropractors and other healthcare professionals are desired by patients and are associated with patients' favourable perception of chiropractic [20, 33]. It may also facilitate inter-professional relations and increase synergies between professions, reducing barriers to collaboration and positively impacting patient outcomes [62, 63]. Participants in this study believed that integration of chiropractors into multidisciplinary settings, improving safe data-sharing technology to enable co-management, and allowing access to critical health-record information between healthcare professionals may facilitate better communication and collaboration for the benefit of the

older patient and improve interprofessional respect, increase patient safety, decrease individual suffering, and helping older patients maintain function and autonomy in old age. Despite this, all participants had a persistent feeling of not being entirely accepted by other healthcare professionals–a sense of being outside, looking in. This has been addressed in different settings and calls for cultivating positive professional attributes, fully embracing evidence-based care and adhering to the standards of other mainstream AHP, for chiropractic care to move towards being fully accepted by mainstream healthcare professionals in the future [27, 64, 65].

## Methodological considerations

We used a purposive sampling of chiropractors in four European countries known to provide care and frequently manage older patients with MSK complaints. Although we aimed for an equal gender distribution, the majority of participants were female (57%), despite chiropractic being a male-dominated profession in the Netherlands (58%), personal communication Monique van der Marck), Norway (64.2%) [66] and Sweden (63%) [67]. Conversely, in the UK, gender distribution among chiropractors is equal [68]. Consequently, our sample may suffer from selection bias and, thus, may not represent all viewpoints of all chiropractors across Europe. Nevertheless, the results of this study give insights regarding chiropractors' experiences in the care and management of older patients with MSK complaints with a focus on organisational challenges, integration, and collaboration among health professionals. Given the wealth of information gathered during the in-depth individual interviews, we strategically divided the material to streamline the presentation findings, facilitating a clearer understanding. While our decision to categorise the data undoubtedly helped illuminate the data, it is important to acknowledge that alternative compartmentalisation strategies might have yielded additional insight and conclusion. However, it is noteworthy that analyses regarding the treatment and management of this patient population will be presented in a subsequent paper.

Participants could choose to perform the interview in English, Swedish, or Norwegian as the lead author (CB) is fluent in all three languages, enabling participants to express themselves more freely and, simultaneously, catch nuances that may otherwise be lost. Participants from the Netherlands could only perform the interview in English, however, after studying chiropractic abroad, all Dutch chiropractors participating in this study considered themselves fluent in English. The pre-existing knowledge of some researchers (chiropractors) in this study may be considered both a weakness and a strength. However, by presenting a robust and transparent methodology encompassing all study phases, the findings' dependability, confirmability, transferability and authenticity were established; thereby, the overall trustworthiness of the results was ensured [69]. Finally, the authors representing a variety of professions, several with extensive clinical experiences, further contributed to minimising bias caused by a preconceived understanding of the topic and cultivating a multifaceted discussion of the results. To guide the reporting of the results, the COnsolidated criteria for REporting Qualitative research (COREQ) checklist [70] was used (S2 Appendix).

## Conclusion

Chiropractors from Great Britain, Norway, the Netherlands, and Sweden emphasised that they perceive to be well-positioned to fill an identified gap regarding the care and management of older adults with MSK complaints. Several barriers to the collaboration and integration of chiropractic into the public healthcare system were identified, and chiropractors saw themselves as an underutilised resource. The findings of the explorative study highlight the desire to foster collaboration among healthcare professionals and incorporate chiropractic services into the national public healthcare system. Integration of chiropractors as AHPs was perceived to

enhance the coordination of care and promote patient-centred healthcare for older adults, potentially resulting in improved outcomes.

## Supporting information

**S1 Appendix. Interview guide.**
(PDF)

**S2 Appendix. COREQ (Consolidated criteria for Reporting Qualitative research) checklist.**
(PDF)

## Acknowledgments

The authors wish to thank all participating chiropractors who generously shared their experiences with us. Special thanks to Sabina Lundberg for transcribing most of the interviews.

## Author Contributions

**Conceptualization:** Cecilia Bergström, Iben Axén, Jan Hartvigsen, Sidney Rubinstein, Margareta Persson.

**Data curation:** Cecilia Bergström, Margareta Persson.

**Formal analysis:** Cecilia Bergström, Margareta Persson.

**Funding acquisition:** Cecilia Bergström.

**Investigation:** Cecilia Bergström, Margareta Persson.

**Methodology:** Cecilia Bergström, Iben Axén, Jan Hartvigsen, Sidney Rubinstein, Margareta Persson.

**Project administration:** Cecilia Bergström, Iben Axén.

**Resources:** Cecilia Bergström, Iben Axén, Jonathan Field, Monique van der Marck, Dave Newell, Sidney Rubinstein, Annemarie de Zoete.

**Supervision:** Cecilia Bergström.

**Validation:** Cecilia Bergström, Iben Axén, Jonathan Field, Jan Hartvigsen, Monique van der Marck, Dave Newell, Sidney Rubinstein, Annemarie de Zoete, Margareta Persson.

**Visualization:** Cecilia Bergström, Margareta Persson.

**Writing – original draft:** Cecilia Bergström, Margareta Persson.

**Writing – review & editing:** Cecilia Bergström, Iben Axén, Jonathan Field, Jan Hartvigsen, Monique van der Marck, Dave Newell, Sidney Rubinstein, Annemarie de Zoete, Margareta Persson.

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
