## [Decision Letter · Decision Letter 0]

7 Feb 2024

PONE-D-23-43815The chiropractors’ dilemma in caring for older patients with musculoskeletal complaints: collaborate, integrate, coexist, or separate?PLOS ONE

Dear Dr. Bergström,

Thank you for submitting your manuscript to PLOS ONE. After careful consideration, we feel that it has merit but does not fully meet PLOS ONE’s publication criteria as it currently stands. Therefore, we invite you to submit a revised version of the manuscript that addresses the points raised during the review process.

We look forward to receiving your revised manuscript.

Kind regards,

Holakoo Mohsenifar

Academic Editor

PLOS ONE

Journal Requirements:

Reviewers' comments:

Reviewer's Responses to Questions

**Comments to the Author**

1. Is the manuscript technically sound, and do the data support the conclusions?

Reviewer #1: Partly

Reviewer #2: Partly

2. Has the statistical analysis been performed appropriately and rigorously? 

Reviewer #1: N/A

Reviewer #2: N/A

3. Have the authors made all data underlying the findings in their manuscript fully available?

Reviewer #1: Yes

Reviewer #2: Yes

4. Is the manuscript presented in an intelligible fashion and written in standard English?

Reviewer #1: Yes

Reviewer #2: Yes

5. Review Comments to the Author

Reviewer #1: Thank you for the opportunity to review this paper. I hope my comments and suggestions will be helpful to strengthen the paper.

1. Title: I rarely comment on a title, but the phrase after the colon "collaborate, integrate, coexist, or separate?" was the main reason I agreed to review the paper. Unfortunately I was disappointed because these topics were not really addressed to any significant degree in the paper. Perhaps a title that better reflects the text would be more appropriate?

2. sampling/recruitment: In the Design section and again in the methodological considerations section, the authors state that they used purposeful sampling. However, in the recruitment and participants section, the authors write that they recruited a "convenience sample". Purposive sampling is a much stronger choice and if used, more details about how this was implemented (i.e., was was considered in making choices re who to interview) should be included in the recruitment section. Currently it is not at all clear how the participants were selected. Similarly, a justification of the four countries is not provided -- why these countries and not others?

3. sample size -- no justification was provided for why 24 chiropractors were invited to participate and/or why when only 21 accepted, additional chiropractors were not invited to participate.

4. Participant demographics (Table 1) -- It would be helpful to know how representative the participants are of the chiropractors in their respective countries. For example, 57% of the participants were women; in some countries the chiropractor profession is male dominated. How des this compare with the sex balance of chiropractors in the four countries from which the participants were sampled? This information would be helpful in assessing the transferability of the findings. I would also have found it helpful to know what proportion of the participants were currently practicing in a private vs public (or both) setting.

5. analysis: The description of the qualitative analysis procedure was excellent, but I was a bit under-whelmed by the findings as described in the paper. For example:

a. all the participant data appear to have been analyzed as a single data set regardless of the country. Given the different contextual factors in each country (not the least of which that in the Netherlands chiropractors are not regulated), it is not clear why there was no attempt to explore similarities and differences of the barriers and facilitators across participants from different countries.

b. given what is likely a very rich data set, I was a bit disappointed in the depth of the qualitative analysis. As summarized in the conclusion "chiropractors .... emphasized that are well positioned to fill an identified gap regarding care and management of older adults with MSK complaints...... chiropractors saw themselves as an underutilized resource." Despite sections on barriers and facilitators (money and money), there was very little new insight offered. And the subject matter (MSK in the elderly) was not a focus of any of the themes -- what is unique in this patient population vs in others?

6. Discussion The questions of "collaborate, integrate, coexist or separate" in the title where not mentioned at all in the discussion. In fact the vast majority of the discussion was simply an argument about why chiropractors should be used more for supporting elderly patients with MSK concerns and how the data supported this rather than any real discussion of the findings. For example, how are the findings similar or different across countries, regulatory systems, health care systems and funding models? What models seem to be working better/worse? What do the participants think about the options of collaborate, integrate, coexist or separate?

7. Discussion: The study was focused on caring for older patients with MSK issues, but it wasn't clear what was unique or different or specific with respect to the focus on older patients. This was really not mentioned at all in the results or the discussion. How are the barriers and facilitators any different in this patient group compared to other patient groups?

The team has obviously put a lot of effort into this project and likely has a very rich data set. I hope these comments are helpful in drawing out some additional insights that may make the paper more impactful.

Reviewer #2: Thank you very much for the opportunity to review this interesting work. This a timely well-conducted qualitative study that provides interesting perspectives into a very important topic: conservative care of MSK conditions in older adults.

I would like to commend the authors for conducting such study in this much needed area. The suggestions below are intended to facilitate readers' understanding and richness of this manuscript.

1. Methods, Setting: it may be helpful for the reader to have additional information on how is the payment method for chiropractic care in these countries. Especially as this is discussed by participants and part of the study's results.

2. Methods, recruitment, line 124: was there a criteria for selecting chiropractors who "frequently" manage patients 55+ years? In other words, who did the investigators define "frequently"? For example, chiropractors who saw X or more patients over 55 years old per week?

3. Methods, data collection, line 149: suggest including references of the literature that the interview guide was based on. This would support the interview guide content.

4. Methods, data collection, line 150: suggest including additional details on how was the interview guide evaluated. Since only one pilot interview was conducted, detailing the criteria for interview guide evaluation can speak to and support the strength of such guide.

5. Results, table 1: consider including information on participants' practice sector (i.e., public versus private). This is also part of the results and would be helpful to know the number of participants who provided their views for each sector.

6. Results, line 392: please clarify what does "immigration background" mean?

7. Results and discussion regarding participants' perceptions of having more MSK knowledge than the average GP. In the discussion (lines 522-525), authors mention that patients in general health practice are more complex than patients seen by chiropractors (i.e., higher rates of comorbidities, experience longer episodes of LBP, have more sick leave, etc.). But participants have the perception that chiropractors are better suited than other health care professionals to assess and manage these patients. Is this a realistic perception? In other words, if they don't normally see patients as complex as other professions, is this belief that they are better suited actually founded? Or could it be just their perception? Could authors expand on that in the manuscript?

8. Discussion, lines 539-547: in results, participants talked about the challenges of free of charge management for patients with low resources and keeping them engaged with active treatment. I.e., the challenges of offering a lower cost to patients who need it, but at the same time motivating their engagement and active participation in their own care. If possible, could authors expand here too and provide a few suggestions from the literature of how this could potentially be approached?

9. Discussion, lines 559-561: this was an interesting statement. There are studies reporting positive views of other health care professionals towards chiropractors. Does that mean other professionals' views don't necessarily translate into action and referrals to chiropractors? Or could it be a perception unique to the study participants? Are there any studies regarding referrals to chiropractors from other healthcare professionals that could be used to expand the discussion a bit here too?

10. Methodological considerations (or in methods): suggest clearly describing that the interview transcripts were not shared with the participants. This is in the COREQ checklist, but should also be explicitly reported in the manuscript.

11. Conclusion, first sentence: Since this is a qualitative study on chiropractor's perceptions, suggest considering the following re-wording: Chiropractors from Great Britain, Norway, the Netherlands, and Sweden emphasised that they perceive to be well-positioned to fill an identified gap...

12. Supplemental material 1, interview guide: The interview guide includes questions about chiropractors education and training to manage this specific population, however this is not reported in the manuscript. Could this potentially provide a rationale for participants' perception of having higher competence in assessing and managing older patients with MSK complaints?

6. PLOS authors have the option to publish the peer review history of their article (what does this mean?). If published, this will include your full peer review and any attached files.

Reviewer #1: No

Reviewer #2: No

---

## [Author Response · Author response to Decision Letter 0]

28 Mar 2024

We would like to express our gratitude to the reviewers for their insightful feedback and suggestions on our manuscript. We are glad to have the opportunity to revise and resubmit our work.

Please find attached the file "Response to Reviewers," which contains the revision notes for PONE-D-23-43815 titled, "The chiropractors' dilemma in caring for older patients with musculoskeletal complaints: Collaborate, integrate, coexist, or separate?" The changes made in the manuscript have been highlighted in yellow for your convenience (Revised manuscript with track changes).

---

## [Editor Report · Decision Letter 1]

8 Apr 2024

The chiropractors’ dilemma in caring for older patients with musculoskeletal complaints: Collaborate, integrate, coexist, or separate?

PONE-D-23-43815R1

Dear Dr. Cecilia Bergström,

We’re pleased to inform you that your manuscript has been judged scientifically suitable for publication and will be formally accepted for publication once it meets all outstanding technical requirements.

Kind regards,

Holakoo Mohsenifar

Academic Editor

PLOS ONE